# Systematic Review: Targeted Molecular Imaging of Angiogenesis and Its Mediators in Rheumatoid Arthritis

**DOI:** 10.3390/ijms23137071

**Published:** 2022-06-25

**Authors:** Fatemeh Khodadust, Aiarpi Ezdoglian, Maarten M. Steinz, Judy R. van Beijnum, Gerben J. C. Zwezerijnen, Gerrit Jansen, Sander W. Tas, Conny J. van der Laken

**Affiliations:** 1Department of Rheumatology and Clinical Immunology, Amsterdam University Medical Center, Location VUmc, De Boelelaan 1117, 1081 HV Amsterdam, The Netherlands; f.khodadustvaskasi@amsterdamumc.nl (F.K.); a.ezdoglian@amsterdamumc.nl (A.E.); m.m.steinz@amsterdamumc.nl (M.M.S.); g.jansen@amsterdamumc.nl (G.J.); 2Angiogenesis Laboratory, Department of Medical Oncology, Amsterdam University Medical Center, Location VUmc, De Boelelaan 1117, 1081 HV Amsterdam, The Netherlands; j.vanbeijnum01@gmail.com; 3Department of Radiology and Nuclear Medicine, Amsterdam University Medical Center, Location VUmc, De Boelelaan 1117, 1081 HV Amsterdam, The Netherlands; g.zwezerijnen@amsterdamumc.nl; 4Department of Rheumatology and Clinical Immunology, Amsterdam University Medical Center, Location AMC, Mijbergdreef 9, 1105 AZ Amsterdam, The Netherlands; s.w.tas@amsterdamumc.nl

**Keywords:** rheumatoid arthritis, angiogenesis, molecular imaging, PET, SPECT, scintigraphy, hybrid imaging

## Abstract

Extensive angiogenesis is a characteristic feature in the synovial tissue of rheumatoid arthritis (RA) from a very early stage of the disease onward and constitutes a crucial event for the development of the proliferative synovium. This process is markedly intensified in patients with prolonged disease duration, high disease activity, disease severity, and significant inflammatory cell infiltration. Angiogenesis is therefore an interesting target for the development of new therapeutic approaches as well as disease monitoring strategies in RA. To this end, nuclear imaging modalities represent valuable non-invasive tools that can selectively target molecular markers of angiogenesis and accurately and quantitatively track molecular changes in multiple joints simultaneously. This systematic review summarizes the imaging markers used for single photon emission computed tomography (SPECT) and/or positron emission tomography (PET) approaches, targeting pathways and mediators involved in synovial neo-angiogenesis in RA.

## 1. Introduction

Rheumatoid arthritis (RA) is a chronic systemic autoimmune disease with an approximate prevalence of 1% of the adult population worldwide. This debilitating heterogeneous disease is characterized by the onset of synovial angiogenesis and inflammation, which eventually leads to the hyperplasia of the synovial lining and joint destruction [1,2,3]. Continuous vascular activity is found to promote cartilage and bone destruction, even when patients are in clinical remission [4].

The healthy synovium is characterized by a unique lining/sublining architecture. There is a delicate intimal lining composed of 1–2 layers of macrophage-like (type A) and fibroblast-like (type B) synoviocytes. Upon inflammatory triggering, the synovium undergoes a transitory pre-vascular stage in which the lining layer expands into sublining through fibroblast growth and extracellular matrix deposition (Figure 1) [5]. Simultaneously, the sublining is occupied by inflammatory cells (macrophages, T cells, and B cells). These events result in synovial tissue proliferation, called pannus, which acts like a local tumor, causing bone erosions observed in RA. In response to the increased requirement of oxygen and nutrients caused by the overpopulation of cells in the synovium, the vascular stage rapidly arises. This stage usually takes place at the time of RA clinical diagnosis. The increased density and permeability of blood vessels boosts synovitis by allowing immune cells to emigrate from the blood into the inflamed synovium, which in turn favors chronicity [3,6,7].

Angiogenesis is a complex multifaceted process that is defined as the formation of new blood vessels from the pre-existing vasculature via sprouting or remodeling. This is distinct from vasculogenesis and neovascularization, which refer to the de novo production of vascular networks from endothelial precursor cells [8,9]. Angiogenesis is involved in several physiological and pathological conditions including various inflammatory diseases such as RA, psoriasis, and atherosclerosis as well as cancer [10]. Under normal conditions, angiogenesis is a self-controlled and self-limiting process with balance in pro/anti-angiogenic factors [7,11,12,13]. In pathological angiogenesis, however, this balance is impaired and causes continuous endothelial cell (EC) activation, basement membrane rupture and migration, followed up by EC expansion, lumen formation and, finally, the formation of new blood vessels [14,15]. Here, the angiogenic process offers a variety of target structures for diagnostic tools and therapeutic interventions [6].

In RA, angiogenesis mainly relies on the dynamic, temporally, and spatially interaction of vascular ECs, fibroblasts, macrophages, and the extracellular matrix (Figure 1) [7,11,12]. Synovial fibroblasts and activated immune cells are responsible for the production of a large number of inflammatory cytokines, which are believed to play a crucial role in the development and progression of RA. Among the pro-inflammatory cytokines, TNF-α, IL-1, IL-6, and IL-8 are the principal cytokines that regulate the formation of other inflammatory mediators in the synovial tissue. These cytokines exert pro-angiogenic effects in RA both through their direct effect on ECs (meaning the response of ECs to the cytokine results in new vessel formation) as well as their indirect effect (by subsequent release of factors that induce endothelial cells angiogenesis) in the inflamed RA synovium producing pro-angiogenic factors including vascular EC growth factor (VEGF) [10,16,17]. VEGF and its receptors are one of the most extensively studied and fundamental players in the regulation of EC proliferation, angiogenesis, and capillary hyper-permeability. The pro-inflammatory cytokine networks stimulate synovial fibroblasts and other cells to release VEGF and other growth factors. Elevated levels of VEGF have been found in the plasma and synovial fluid of RA patients [18]. Moreover, correlations between VEGF serum level and inflammatory parameters of RA including erythrocyte sedimentation rate (ESR), C-reactive protein (CRP), and disease activity score 28 (DAS28) have been reported [19,20,21,22,23,24]. In the synovial angiogenesis cascade of events, other important mediators have also been recognized, which mostly involve matrix-degrading proteases, cellular adhesion receptors as well as basal membrane and interstitial matrix macromolecules (Figure 1) [16].

At present, there is no cure for RA and current treatment strategies mainly attempt to achieve remission or a low disease activity state (LDAS) as much as possible [1,2]. Three classes of therapeutic agents are commonly used in RA treatment: classical disease-modifying anti-rheumatic drugs (cDMARDs) including steroids, targeted synthetic (tDMARDs) and biological (bDMARDs), which can clinically mitigate the severity of RA, attenuate disease progression, and inhibit subsequent joint damage. None of these therapies specifically target angiogenesis in RA. However, several of these c/t/bDMARDs have been reported to partially impact angiogenesis at different levels [3,25,26,27,28,29,30,31,32,33]. For instance, blocking of the JAK/STAT-pathway by tofacitinib inhibited the formation of new synovial vessels [31]. Furthermore, some clinical studies and preclinical models of arthritis have revealed the benefits of anti-angiogenic therapies [31,33]. Notably, the proposed anti-angiogenic agents for RA were originally developed for cancer therapy and exploit some biological analogies between cancers and synovial pannus [7,34]. Due to the central position of angiogenesis in RA synovitis, therapeutic targeting of synovial angiogenesis by novel anti-angiogenic drugs may offer new opportunities for RA treatment.

The optimal management of RA relies on tools that can support clinical assessment in early and accurate disease detection, the prediction of prognosis, and monitoring of treatment response [35]. Diagnosing RA can be complicated due to the heterogeneous nature of the disease [1,36]. Although no detailed diagnostic criteria exist, the classification criteria of the American College of Rheumatology/European League Against Rheumatism [ACR/EULAR] Rheumatoid Arthritis Classification criteria 2010 [37] serve as a standard format to support clinical diagnosis. These criteria are based on clinical signs and serological assays (autoantibody and acute-phase reactant levels) and incorporate imaging techniques such as conventional radiography, ultrasound (US) and magnetic resonance imaging (MRI) for the detection of synovitis to improve earlier diagnosis and correct classification of patients [1,38,39].

Molecular imaging may offer additional opportunities for early RA disease activity assessment, prediction of treatment response, and monitoring during treatment. Over the past decade, several noninvasive molecular imaging techniques such as conventional planar (2D) scintigraphy and 3-dimensional (3D) single photon emission computed tomography (SPECT), positron emission tomography (PET), and hybrid modalities have been developed and applied for RA diagnostic purposes [40,41]. These modalities rely on the in vivo biodistribution and quantification of the radiotracer binding to a specific molecular target in the body, allowing for exceptional target specificity and sensitivity [42,43]. Traditional 2D scintigraphy and SPECT are imaging techniques that use gamma cameras to detect gamma rays emitted by radionuclides. Unlike planar scintigraphy, SPECT provides 3D tomographic images and yield slices through the body. Typical radionuclides used for planar scintigraphy and SPECT imaging are technetium ^99m^Tc (t_1/2_: 6.0 h), indium ^111^In (t_1/2_: 2.8 days), and iodine ^123^I (t_1/2_: 13.2 h) [44,45]. PET adopted different isotopes (positrons) for imaging and holds certain superiority, resulting in increased popularity both in preclinical and clinical settings. It provides improved imaging quality with higher diagnostic accuracy and sensitivity, allowance for reliable quantification, and mostly a lower radiation burden to patients. Most used positron-emitting radioisotopes clinically and/or pre-clinically include fluorine ^18^F (t_1/2_: 110 min), carbon ^11^C (t_1/2_: 20 min), copper ^61/64^Cu (t_1/2_: 3.3 h and 12.7 h), gallium ^66/68^Ga (t_1/2_: 9.5 and 1.1 h), zirconium ^89^Zr (t_1/2_: 78.4 h), and iodine ^124^I (t_1/2_: 100.2 h) [45,46]. These radionuclides can be conjugated with an appropriate ligand with available binding sites. Depending on the type of radionuclide (e.g., metallic or non-metallic) and/or achieving the desirable biological activity of a targeting vector, the chelator and/or linker (spacer) can be utilized with which the in vivo biological behavior of the radioligand can be significantly affected [47]. The structural components of radiopharmaceuticals discussed below in the context of angiogenesis research and applications commonly comply with the four compartment criteria depicted in Figure 2A [44,48]. Targeting vector can be bound to a radionuclide and ensures accumulation into the target cells. Linkers and chelators can be used to attach the radionuclide and targeting vector. Some radionuclides, particularly radiometals (such as ^99m^Tc, ^66/68^Ga), require chelators to form stable complexes. Linkers or spacers are often optional components and usually are used to improve the pharmacokinetics of a radiotracer and/or reduce the steric interference and retain high binding affinity [44,47,48]. For clinical utility, it is of importance that a radiotracer meets some basic requirements. In this regard, ideal radiotracers should harbor the properties indicated in Figure 2B [49].

Although both SPECT and PET techniques have deep signal penetration and demonstrate high sensitivity in molecular targets/processes imaging, their spatial resolution is not as high as CT or MRI. However, high contrast resolution and the capability of providing functional images make these two imaging modalities highly attractive, and new technological improvements in PET scanners (total body PET) allow for significant improvement in spatial resolution [50,51]. The feasibility of hybrid imaging systems has facilitated the combination of high sensitivity radionuclide imaging with high-resolution CT imaging, allowing for co-localization of molecular images with anatomical images. PET/MR has important (pre)clinical potential by also providing soft tissue morphological imaging. Ultimately, these imaging approaches may contribute to better-individualized health care as physicians are informed by more accurate and quantified molecular signals within specific anatomical structures of interest [42]. The use of molecular imaging tools may facilitate specific drug development and provide guidance in personalized medicine [1,52,53].

It is worth mentioning that molecular imaging modalities, beyond nuclear imaging techniques, also consist of two other non-ionizing main categories: optical and acoustic (ultrasound) imaging. Although optical techniques such as bioluminescence and fluorescence imaging as well as acoustic imaging provide a highly versatile platform for noninvasive in vivo molecular imaging [54,55], they were beyond the scope of this paper. This systematic review is focused on non-invasive in vivo nuclear imaging.

Concerning angiogenesis imaging, computed tomography (CT) and magnetic resonance imaging (MRI) may not be suitable to assess the response to anti-angiogenic treatment since these techniques merely allow for the evaluation of morphological and perfusion parameters, but do not enable the depiction of changes in the vasculature at the molecular level [15,53]. ^18^F-FDG PET/CT is a sensitive technique for evaluating disease activity and treatment response in patients with RA [56]. However, the mechanism underlying ^18^F-FDG uptake is rather unspecific and is largely associated with elevated glucose metabolism in inflamed cells and tissues [57]. In the case of visualizing angiogenesis in early, developing RA for diagnostic purposes and monitoring anti-angiogenetic properties of RA drugs, molecular imaging markers that target synovial angiogenesis are needed.

The objective of this systematic review was to present an overview of all the potential molecular imaging tracers (PET and SPECT/scintigraphy) involved in the RA angiogenic cascade that have been studied in pre-clinical and/or clinical settings in RA.

## 2. Methods

### 2.1. Eligibility Criteria

This systematic review focused on studies investigating the detection of the pathological angiogenesis in RA through molecular imaging techniques. All studies on the application of radiotracers using PET and SPECT/scintigraphy for diagnosis, prognosis, and response assessment to therapy in preclinical and/or clinical stages of RA (related to angiogenesis) were included. No restrictions were placed on the type of studies included as research on exploiting angiogenesis for RA molecular imaging is still in development.

### 2.2. Information Source and Search Strategy

For this systematic review, a literature search was carried out using the following databases without date and language restriction until March 2022: PubMed, EMBASE, and the Cochrane Library. The literature was reviewed following the Preferred Reporting Items for Systematic Reviews and Meta-Analyses (PRISMA) guidelines [58].

Two separate searches were conducted in consultation with the medical information specialists at the University Library VU (Vrije Universiteit Amsterdam); the initial search focused on all radiopharmaceuticals used for imaging in RA by applying the controlled vocabulary search terms and free-text terms (medical subject headings (MeSH) and relevant keywords) based on the following words: PET/SPECT/Scintigraphy imaging and arthritis. An additional search was performed to identify important angiogenesis biomarkers in RA, through which radiopharmaceuticals that may be useful for angiogenesis targeting in RA were selected. The full search strategies for all databases can be found in the Appendix A (RA angiogenesis biomarkers search) and Appendix A (RA radiotracers search).

### 2.3. Study Selection

Both co-reviewers (F.K. and M.K.) independently screened all potentially relevant titles and abstracts for eligibility. The initial evaluation was based on title, keywords, and abstracts of the retrieved studies. If necessary, the full-text article was checked for the eligibility criteria. This evaluation was performed at Rayyan QCRI (Qatar Computing Research Institute) on a platform expediting the initial screening of abstracts and titles; it allows for the transfer of the citations obtained from different databases and provides an individual blinded assessment of articles with the ability to invite an unlimited number of collaborators [59]. Disagreements after an initial blinded evaluation were resolved through discussion. Following consensus, full-text versions of the studies that met the eligibility criteria and those that were not considered eligible were acquired and assessed for the final determination of inclusion.

### 2.4. Data Extraction and Assessment

Data from the studies included were extracted by two authors (F.K. and M.K.). The reviewers independently extracted data for each search (angiogenesis biomarkers and RA radiotracers) and then a third reviewer (A.E.) randomly evaluated the collected data. The initial data extraction for the angiogenesis biomarkers of RA included the purpose of study (exploratory, treatment, diagnosis), diagnostic or therapeutic interventions used, molecular targets, and clinical stage. For radiotracers in RA, we characterized articles according to the type (PET and SPECT/scintigraphy), name of the tracer, and molecular targets.

### 2.5. Search Results

The first literature search that focused on nuclear imaging in RA resulted in 356 studies. The selected articles included case reports, letters, and the investigation of the pharmacokinetics of the tracers as well as tracers in both animal and human trials. In the second search, which focused on angiogenesis targets in RA, 323 potentially related studies were included. This search included studies that explored the role of a biomarker(s)/pathway(s) in the angiogenesis process, or reported the mechanism of action (or target) of a therapeutic or diagnostic approach involved in the angiogenesis in RA in pre-clinical/clinical settings. Next, the final related articles were selected by matching the angiogenesis targets in both searches. The flow diagram of the search and selection process is presented in Figure 3A,B. The final total number of included articles in the results was 39 studies. The radiotracers and accompanying articles were divided based on the angiogenesis targets (Table 1).

## 3. Results and Discussion

ResultsRadiotracers targeting angiogenesis in RA

The PET and SPECT/scintigraphy tracers included in this review targeted different levels of pathological angiogenesis in RA. The radiotracers were categorized based on the type of targets reported to be involved in RA angiogenesis (Figure 1). An overview of the radiotracers is presented in Table 1 and is discussed in detail below.

### 3.1. Environmental Factor (Hypoxia)

Enhanced angiogenic activity in the synovial vasculature of arthritic joints is associated with the onset of a hypoxic environment [10]. Many studies have revealed the connection between VEGF and hypoxia [99,100,101,102]. Excessive leukocyte migration into the RA joint and hyperplasia of the synovial lining cells confers joint microenvironment hypoxia, leading to the activation of the key transcription factor HIF-α (two isoforms, HIF-1α, HIF-2α). This transcription factor induces the expression and secretion of VEGF by macrophages and RA synovial tissue fibroblasts. During hypoxia, a positive feedback between the HIF-1α and VEGF pathways occurs, which regulates continuous angiogenesis. Hypoxia, in turn, can facilitate the expression of many pro-angiogenic and pro-inflammatory genes including the chemokines IL-8 (also known as CXC-chemokine ligand 8 or CXCL8), CC-chemokine ligand 20 (CCL20), and CXCL12 [10,103,104].

So far, radiolabeled azomycins (2-nitroimidazoles) such as ^18^F-FMISO (^18^F-Fluoromisonidazole) [60,61], ^18^F-FAZA (^18^F-fluoroazomycinarabinoside) [61], ^123^I-IAZA (1-(α-D-5-Iodoarabinofuranosyl)-2-nitroimidazole) [64], and more recently ^64/67^Cu-ATSM (copper bis-thiosemicarbazone complexes) [62] have been explored for hypoxia imaging in RA.

The HIF-1α tracers, ^18^F-FMISO and ^18^F-FAZA, are among the commonly clinically used tracers that have been investigated for the identification of tumor hypoxia in malignancies. These nitroimidazole compounds are metabolically trapped in tissues with low oxygen tensions [105]. PET studies using these hypoxia tracers in a glucose-6-phosphate isomerase (GPI)-induced arthritis mouse model revealed an increased uptake in the arthritic joints with comparable arthritis-to-control joint ratios (ratio ^18^F-FMISO: 2.7; ratio ^18^F-FAZA: 2.8). However, ^18^F-FAZA exhibited a greater hydrophilicity and faster tissue clearance, and thus reduced whole-body background compared to ^18^F-FMISO. Histological studies, simultaneous MRI imaging, and clinical swelling score of the inflamed joints correlated well (especially for ^18^F-FMISO) with the uptake of the radiotracers. The utility of these PET tracers has been suggested for the detection of early pathological changes in the initial phases of RA before the onset of cartilage and joint destruction [61].

In another study, ^18^F-FMISO was used to explore the therapeutic potential of two different classes of novel MAPK inhibitors (the highly selective p38 MAPK inhibitor (Skepinone-L) and the dual p38 MAPK/JNK 3 inhibitor (LN 950) in the K/BxN serum transfer model of RA [60]. MAPK activity is associated with hypoxia and is a crucial factor in RA progression through the propagation of angiogenesis and pannus formation. Both inhibitors markedly suppressed arthritic disease activity, but the radiotracer uptake in the ankles of skepinone-L-treated mice did not reflect the clinical parameter of ankle swelling very well. It was concluded that this hypoxia targeting tracer might not be a reliable tool to monitor MAPK inhibition when compared to the parameter of ankle swelling, although the MAPK inhibitors directly influence hypoxia via the modulation of cell proliferation and angiogenesis [60].

The in vivo behavior of another hypoxia radiotracer, ^64/67^Cu-ATSM, which is mainly used in oncology, cardiology, and neurology, has recently been assessed in the collagen-induced arthritis (CIA) mouse model [62]. The ^64/67^Cu-ATSM study data exhibited a correlation between the accumulation level of the tracer and the clinical scores of the arthritic paws. However, when validating the specificity of the targeting hypoxia, hypoxic tissue conditions of the inflamed joints were not confirmed by the analysis of other markers for hypoxia [62].

^123^I-IAZA has also been developed for the detection of tissue hypoxia [63]. Nevertheless, in a clinical study with scintigraphic evidence of arthritic joint hypoxia, the uptake intensity of this radiotracer did not correlate with the disease severity in RA patients [64].

### 3.2. Somatostatin Receptors—Somatostatin

Somatostatin is a natural, widely distributed, small cyclic peptide that inhibits exocrine secretion, cellular proliferation, and cellular differentiation and promotes apoptosis. The effects of somatostatin and its analogues are mediated via five (G-protein-linked) receptors (SST 1, 2, 3, 4, 5). These receptors regulate multiple signal transduction pathways to elicit their inhibitory effects [106]. The modulatory effects on immune response, anti-proliferative, anti-angiogenic, and analgesic properties of somatostatin and its analogues have been explored [107]. Both the activated lymphocytes and inflamed vascular endothelium can express SSTR 2, 3, and 5 in different physiological and pathologic conditions [108].

Some studies have shown that SST may have a functional role in angiogenesis. It has been reported that SST2 is particularly expressed in the angiogenic sprouts of endothelial cells (HUVEC) in vitro, and experimental angiogenesis is inhibited by the synthetic somatostatin analogue (e.g., octreotide) [106,109,110,111]. In RA, somatostatin agonists have been demonstrated to potently abrogate VEGF levels as well as pro-angiogenic activity. Additionally, SST2 is uniquely upregulated during the angiogenic cascade. The receptor exists in two isoforms: SST2A and SST2B. The isotype SST2A is expressed at a high density during inflammatory response in arthritis mouse models. Synovial biopsies from RA patients have demonstrated the presence of SST2A in the synovial venule endothelium as well as on a subset of synovial macrophages [112]. In vitro, somatostatin treatment on RA synoviocyte cells was associated with the downregulation of MMP. Specifically MMP-1, MMP-2, and MMP-9 are known to trigger the angiogenic cascade [113]. These characteristics make somatostatin/SSTs attractive candidates in the context of angiogenesis for both diagnostic and therapeutic use in immune-mediated diseases such as RA [108].

In an original study, Vanhagen et al. reported the use of somatostatin scintigraphy for the visualization of clinically affected joints in RA patients [67]. The results demonstrated that the SST2-preferring analogue ^111^In-pentetreotide (or [^111^In-DTPA-D-Phe1]-octreotide) accumulated in active rheumatoid joints, but not in osteoarthritic joints [67].

In another clinical study in 18 RA patients, a radiolabeled somatostatin analogue (^99m^Tc-EDDA/HYNIC-TOC) was used to evaluate the activity status of joint inflammation. The patients were also scanned after receiving infliximab (anti-TNF-α) therapy [66]. The results showed intense uptake with a symmetric and focal pattern in the hands, predominantly in the carpal, metacarpal, and proximal interphalangeal joints. All patients and all positive joints showed a clinical and scintigraphic improvement after infliximab therapy. No correlation was observed between joint pain or swelling and somatostatin receptor scintigraphy positivity. Some joints that were apparently clinically poorly positive showed high somatostatin uptake in the images (potentially pointing at subclinical disease detection) and vice versa. The targeting was explained to be associated with the overexpression of somatostatin receptors in active phases of the disease, which is characterized by endothelial activation and lymphocyte infiltration in the synovium [66,114].

The ^99m^Tc-depreotide is a ^99m^Tc-labeled somatostatin analogue with a high affinity for subtypes 2, 3, and 5 of somatostatin receptors [68]. In a study, this tracer exhibited good sensitivity to localize in the foci of active osteomyelitis, however, it was not specific and also could detect recent osteonecrosis and RA. ^99m^Tc-depreotide scintigraphy was proposed as a complementary method for the investigation of bone infection and inflammation, particularly osteomyelitis and cases of recent osteonecrosis and RA. The authors postulated that since these particular receptors are overexpressed on the surface of activated leucocytes, the detection of the inflammation by this tracer can be associated with the increased vascular supply and permeability, along with considerable leucocyte infiltration [68].

Shamim et al. conducted a clinical study for a comparative evaluation of the conventional MDP bone scan and ^68^Ga-labeled SST analogue DOTA-NOC PET/CT in the articular manifestation of RA [65]. The radiolabeled SST analogue did not exhibit superiority over the MDP scan scintigraphy for the detection of the clinically diagnosed joints for RA. SSTR PET/CT and MDP scans were almost equally able to detect the RA affected joints with a detection rate of 77% and 80% of clinically positive joints, respectively. Nevertheless, this radiotracer can be considered as a theranostic tool for RA patients [65].

### 3.3. Matrix Metalloproteases

Matrix metalloproteinases (MMPs) are a subfamily of zinc-dependent enzymes and on the basis of their substrates and the organization of their structural domains are classified into collagenases, gelatinases, stromelysins, matrilysins, membrane-type (MT)-MMPs, and other MMPs [115]. They can be secreted by various cell types such as fibroblasts, vascular smooth muscle, and leukocytes and contribute to tissue remodeling during various physiological and pathological processes [115,116]. MMPs promote EC migration and tube formation by proteolytically remodeling the basement membrane [117]. Their distribution is restricted in normal joint tissue; however, in response to inflammatory cytokines and the production of growth factors, MMPs are expressed in the synovium and cartilage. In RA, MMP-3 is mainly produced by synovial lining layer cells and by ECs, while MMP-9 (gelatinase B) and MMP-13 (collagenase 3) proteins are expressed mainly by ECs, fibroblasts, chondrocytes, and leukocytes within the inflamed synovium. Membrane type 1-MMP (MT1-MMP) mRNA has been detected in ECs and fibroblasts [118,119]. MMPs contribute to angiogenesis by breaking down the basement membrane and other ECM components, not only allowing ECs to detach and migrate into new tissue, but also by releasing ECM-bound proangiogenic factors (bFGF, VEGF, and TGFβ) [120,121,122]. 

#### MMP-2/-9/-13

Among the MMPs, MMP-9 and MMP-13 have received specific focus for their proactive role in the angiogenic process in the arthritic joint. Research on the association of MMP-2 and MMP-9 and MMP-1 and MMP-13 with VEGF in the synovial fluid of RA patients has shown that only the MMP-9 and MMP-13 levels were significantly correlated with the VEGF levels [10,123]. Moreover, a strong positive association was observed between MMP-13 with MMP-9 and urokinase plasminogen activator (uPA). Among the various MMPs, MMP-9 and MMP-13 cooperate in joint degradation, and the expression levels of both factors were markedly elevated in the sera and joints of RA patients when compared to osteoarthritis (OA) and associated with disease progression and severity. The potential role of uPA in angiogenesis through the activation of MMP-13 has also been outlined [124]. Based on the association between MMP-9 and VEGF in RA patients [123] and the reduced severity of antibody-induced arthritis in MMP-9 knockout mice [125], it is assumed that the inhibition of MMP-9 can attenuate angiogenesis in the pannus and inhibit the activation of proMMP-13 (inactive MMP-13) in the arthritic joint [124]. The membrane-type matrix metalloproteinases (MT-MMPs) have shown to cleave and activate other MMPs such as MMP-2 and MMP-13. In MT1-MMP (MMP-14) knockout mice, the severe phenotype with impaired angiogenesis has been suggested to occur due to the lack of the activation of MMP-13 and MMP-9 [126].

Breyholz et al. evaluated a series of barbiturate derivatives (pyrimidine-2,4,6-triones (RO 28-2653)) as a new class of potential selective MMP-inhibitor (MMPI) radiotracers by radiolabeling them with iodine-125 [127]. Indeed, RO 28-2653 elicited antiangiogenic efficacy due to the inhibitory activities on MMP-2, MMP-9, and MMP-14. These observations supported the approach of using barbiturates as a radiotracer to visualize angiogenesis [127,128]. Structural modifications have been performed in the barbiturate derivative by converting it to a more hydrophilic form using a mini polyethylene glycol (PEG) linker. Preliminary biodistribution studies with one of these modified structures (^18^F-26) indicated no tissue-specific accumulation in wild-type mice [128]. In an extension to this work, a new series of fluorinated pyrimidine 2,4,6-trione-based MMPIs derivates (e.g., ^18^F-30) was identified as potent inhibitors of MMP-2, -8, -9, and -13. The radiotracer ^18^F-30 exhibited excellent stability in vitro and rapid clearance, as shown by in vivo biodistribution studies in wild-type mice [69]. The approach of barbiturate-based MMP-targeted radiotracers was proposed to be a viable non-invasive approach of in vivo PET imaging of MMP-2 and MMP-9-associated diseases including arthritis.

More recently, N,N′-bis(4-fluoro-3-methylbenzyl)pyrimidine-4,6-dicarboxamide-based selective and potent MMP-13 inhibitors have been successfully synthesized and radiolabeled with ^11^C, ^18^F, and ^68^Ga [70]. Among them, the ^11^C-methylated compound 2d represented the most lipophilic radiotracer with predominant hepatobiliary excretion and remaining tracer accumulation over time in the liver. In contrast, gallium-labeled derivatives revealed improved clearance characteristics achieved by introducing a more hydrophilic DOTA moiety to the conjugate. Therefore, preclinical evaluation of these PET radiotracers in murine disease models expressing upregulated levels of MMP-13 (e.g., breast cancer and arthritis) has been proposed [70].

### 3.4. Extracellular Matrix Proteins

Extracellular matrix (ECM) components including matrix molecules within the EC basement membrane such as type I collagen, laminin, fibronectin, vitronectin, tenascin, and various proteoglycans have been implicated in EC migration during angiogenesis [16]. The ECM controls EC activities by diverse mechanisms ranging from cell anchorage, integrin-mediated activation, and signaling to the binding, release, and activation of soluble growth factors and alteration in the supramolecular matrix architecture [129].

#### Extra Domain A (ED-A)

Fibronectin (FN) is an ECM protein that binds to integrins and to other ECM proteins. Fibronectin polymorphic variations result in the expression of three different domains [130]. Extra domain-A and -B (ED-A/-B) isoforms are recognized as important angiogenic markers that are synthesized, secreted, and deposited to the ECM structures from different cell types residing in the synovial lining layer. Studies have been conducted to determine the functional roles of the ED-A and ED-B domains of FN. The ED-A domain is involved in various processes including cell adhesion, myofibroblast differentiation, wound healing, cell cycle progression, and mitogenic signal transduction [131,132,133]. High expression levels of both isoforms have been detected in human pannus tissues on neovascular sites [132,134,135,136,137]. ED-A/-B have a restricted expression pattern in normal tissues, which is in contrast to fetal and transformed human tissues. Together, these findings have established fibronectin splicing variants as attractive targets for molecular imaging-based detection and monitoring/predicting the outcome of anti-angiogenic therapies [46].

Antibody-based pharmaco-delivery strategies, mainly studied in oncological settings, have also recently been proposed for use in RA. In this regard, the targeting properties and biodistribution of an ED-A binding antibody fragment (F8) conjugated with IL-10 (anti-inflammatory cytokine) were explored in preclinical and clinical settings via PET-guided imaging [138]. Iodine-124 labeled F8−IL10 accumulated not only in the clinically inflamed joints of patients, but also in the subclinically affected joints. These results were corroborated in an arthritis model in rats. Furthermore, this PET tracer showed rapid blood clearance and increased radioactivity in the spleen and liver. The radiotracers’ pharmacokinetic comparison in healthy and arthritic rats revealed higher uptake levels in the liver and spleen of the arthritic model. This indicates that the high uptake in these organs may be ED-A specific and seems to be associated with systemic inflammation. The encouraging results from F8-IL10 therapy studies [138] may hold promise for using this PET tracer to select patients who may benefit from this therapeutic approach [72].

### 3.5. Adhesion Molecules

Adhesion molecules are generally classified into five groups based on their molecular structure and mode of interaction [139] and members of each group have been detected in angiogenic blood vessels [140]. These groups include: (1) integrins (e.g., α*_V_*β_3_, α*_V_*β_5_, α_5_β_1_); (2) selectins (E/P-selectins); (3) proteins of the immunoglobulin superfamily (IgSF) including nectins and others such as mucins (e.g., VCAM-1, ICAM-1, JAM-A/C, Lewis^y^/H and MUC18); (4) cadherins [16,139,141]; and (5) proteins with enzymatic properties such as vascular adhesion protein 1 (VAP-1) [139,142] also play significant roles in perpetuating the inflammatory response and promoting angiogenesis. In fact, angiogenesis or vascular remodeling depends on both the growth factor stimulation and cell adhesion events [143]. The expression and function of vascular cell adhesion molecules (CAMs) are dynamically regulated following tissue alterations induced by the surrounding stromal changes. This property enables ECs to leave the quiescent state and re-enter the angiogenic cascade [144].

#### 3.5.1. VAP-1

Vascular adhesion protein-1 (VAP-1/SSAO) is mainly expressed on vascular ECs, on smooth muscle cells, and on adipocytes. It is practically absent from the endothelial surface of normal tissues, while upon inflammation, it is rapidly translocated from the intracellular storage granules to the EC surface. VAP-1 contributes to several steps in the extravasation cascade and controls the trafficking of lymphocytes, granulocytes, and monocytes to sites of inflammation. Aside from being an adhesion molecule, it also harbors enzymatic properties [145,146]. MMPs can cleave VAP-1 and generate a soluble form of VAP-1 (designated sVAP-1 or semicarbazide sensitive amine oxidase, SSAO). Through its enzymatic action, VAP-1 oxidizes primary amines and produces hydrogen peroxide, aldehyde and ammonium, which can function as potent inflammatory mediators, leading to the upregulation of other adhesion molecules such as E- and P-selectin, ICAM-1, and VCAM-1. This enzymatic activity is indispensable for leukocyte extravasation through the endothelium. These unique features distinguish VAP-1 from other conventional adhesion molecules. Attempts have been made to elucidate the role of VAP-1 in the pathogenesis of arthritis in vivo, and for this purpose, several SSAO inhibitors of different classes have been developed and applied as potential therapeutic agents for both experimental and clinical use in arthritis [147,148,149,150,151,152]. However, it is still unclear whether the proangiogenic functions of VAP-1 merely rely on its ability to recruit VEGF-producing myeloid cells in different settings, or whether endothelial SSAO activity directly promotes neo-angiogenesis [146,151]. Interestingly, Siglec-9 and -10, both sialic acid-binding immunoglobulin-like lectins, have been identified as counter receptors for VAP-1 on B cells and granulocytes/monocytes, respectively [145,153,154].

Siglec-9 is expressed on all peripheral blood granulocytes where it can bind to the enzymatic groove of VAP-1. Siglec-9 expression is rapidly upregulated on the leukocyte surface after inflammatory stimuli such as TNF-α and lipopolysaccharide (LPS). TNF-α exposure increased Siglec-9-mediated granulocyte binding to the vasculature in the inflamed human synovium [145]. In order to analyze whether a Siglec-9 peptide could be used for in vivo inflammation imaging, it was radiolabeled with 68-gallium and its efficacy in the specific targeting of VAP-1 activated vasculature could be confirmed [145]. Gallium-68-labeled Siglec-9 has been explored for the assessment of synovitis in a rabbit model of synovitis (phytohemagglutinin-induced inflammation model) and the result was compared with ^18^F-FDG [74]. With ^68^Ga-DOTA-Siglec-9, the mild synovitis observed in this model was detected comparably to ^18^F-FDG (inflamed-to-control joint ratios were 1.2  ±  0.14 and 1.5  ±  0.4, respectively). Of note, VAP-1-positive vessels were found in the synovium, both in the PET images and immunohistochemistry (IHC) of the animal joints, which was considered to be due to the systemic response to the chemically-induced inflammation. Furthermore, it seems that there is a distinct VAP-1 expression pattern between humans and preclinical models of arthritis. In the arthritic animals, the mild synovitis induction and the level of vascular VAP-1 expression was lower when compared to that of the RA patients. In humans, VAP-1 expression on the vessel surface was not found in healthy synovial tissue [74]. 

The aforementioned observations in the rabbit synovitis model were consistent with the results of SPECT imaging using ^123^I-labeled fully human anti-VAP-1 monoclonal antibody BTT-1023 [106]. BTT-1023 also binds to the VAP-1 of other primates and rabbits but does not cross-react with rats or mice [75]. In this study, the distribution and pharmacokinetics were also assessed by PET/CT (through radiolabeling of the mAb with ^124^I) in healthy rabbits up to 72 h after injection. ^124^I-radioactivity accumulated in the thyroid gland and the liver. The tracer’s accumulation in these organs was attributed to the deiodination of the radiotracer and the previously mentioned different VAP-1 expression pattern in the animal models, respectively. All of the other organs had a low tracer uptake. The study revealed an inflammation-to-control joint uptake ratio of 1.2 ± 0.1 in the rabbits with arthritis. It was also mentioned that the estimated mean effective dose for a 70-kg man would be rather high for the PET version of this tracer as the estimated injected radioactivity of iodine-124 ideally needs to be higher than that of other PET radionuclides (as only 25% of the ^124^I decay is by positron emission). However, the estimated human radiation burden can be controlled by applying an adequate shielding and dose regimen [75].

Recently, the first-in-human study of ^68^Ga-DOTA-Siglec-9 was performed in order to detect early stage arthritis [73]. ^68^Ga-DOTA-Siglec-9 PET/CT and ^18^F-FDG PET/CT were performed in one patient with RA. The in vivo stability test of the tracers in plasma by HPLC analysis revealed, within 10 min, several peaks representing metabolites other than the radiotracer. The tracer pharmacokinetics performed in healthy men exhibited rapid renal clearance due to the hydrophilic and small size of the peptide [73]. A 30 min dynamic PET/CT scan of the hands of the RA patient showed uptake in the arthritic phalangeal joints. However, the observed SUVs of ^68^Ga-DOTA-Siglec-9 (0.7–1.05) were slightly lower than those of ^18^F-FDG (1.08–1.46). Although this radiotracer harbors promising properties, further optimization is required prior to application in large-scale clinical studies [73].

#### 3.5.2. E-Selectin

Selectins are transmembrane proteins expressed on blood platelets and Weibel–Palade bodies of ECs (P-selectins), leukocytes (L-selectins), and ECs (E- and P-selectins). Selectins are involved in the recruitment of leukocytes from the bloodstream in a cascade-like manner into the inflamed tissue. E-selectin is not expressed under basal conditions [155]. Antibodies to soluble E-selectin block the chemotactic activity of rheumatoid synovial fluid on the ECs and its angiogenic activity both in vitro and in vivo [156]. Soluble E-selectin–induced angiogenesis is predominantly mediated through the SRC-phosphoinositide 3-kinases (PI3K) pathway [155], an important signaling pathway in the angiogenesis process [157]. Of note, a high expression of the inhibitor of DNA binding/differentiation (Id) in ECs within the RA synovial tissues was observed and in functional studies, the overexpression of Id promoted the expression of E-selectin [158]. The synovial fluid and serum levels of soluble E-selectin—together with bFGF—were significantly elevated in RA compared to the OA patients and the controls [159].

The murine 1.2B6 monoclonal antibody (mAb) has been found to selectively target E-selectin, and a Fab fragment of 1.2B6 labeled with ^99m^Tc was compared with ^111^In-labeled 1.2B6 F(ab′)_2_ in RA patients using conventional bone scanning [77]. ^111^In-labeled 1.2B6 was previously validated in experimental arthritis in pigs and subsequently tested in clinical studies in RA. In preliminary studies in RA patients, the labeled 1.2B6 tracer showed higher specificity and better image contrast compared to HIG (nonspecific immunoglobulin tracer) coupled with either ^111^In or with ^99m^Tc. However, the ^111^In-variant was not considered the tracer of choice due to practical reasons such as a higher radiation burden compared to Tc99m for further investigations [77,160]. Subsequently, ^99m^Tc-labeled 1.2B6 Fab (^99m^Tc-Fab) was demonstrated to be as effective as ^111^In-F(ab′)_2_. In fact, ^99m^Tc-Fab can be used in a one-day protocol as the scan outcome appeared to be equivalent at 4 h compared with the ^111^In-F(ab′)_2_ scan outcome at 24 h in patients with RA. In addition, ^99m^Tc Fab demonstrated more specific targeting of active joint inflammation compared with ^99m^Tc-HDP [77].

Finally, nano-SPECT/CT imaging using ^111^In-1.2B6 was performed in a chimeric SCID mouse model as a novel tool for the pre-clinical development of radiopharmaceutical and delivery agents targeting human synovial tissue in vivo [76]. In this study, the uptake of this radiotracer was clearly visible in the human synovial tissue xenograft in the SCID mice at 24 h [76].

#### 3.5.3. Integrins

Integrins are heterodimeric adhesion glycoproteins and comprise two different subunits, an α subunit and a β subunit. These subunits have particular ECM protein binding sites to regulate essential cellular survival, motility, migration, inflamed responses and invasion [161]. α_v_β_3_ and α_5_β_1_, Arg-Gly-Asp (RGD) binding integrins, have been described in RA synovial tissue for their involvement in angiogenesis by promoting EC migration and survival, and their expression in fibroblasts, ECs, and synovial infiltrated cells [161]. The RGD sequence is a common cell recognition motif within the sequence of ECM proteins (e.g., vitronectin, fibronectin, fibrinogen etc.) and is required for interaction with integrins [162]. α_v_β_3_ and α_5_β_1_ are recognized as fibronectin receptors. α_v_β_3_ can also bind to vitronectin, fibronectin, osteopontin, and bone sialoprotein [162,163,164].

α_v_β_3_ and α_5_β_1_ integrins can exert their angiogenic effects by regulating MMP expression. Growth factors, cytokines, hormones, and the composition of the ECM are able to activate integrins and alter their binding affinity. Once activated, several signaling cascades such as focal adhesion kinase (FAK), the mitogen-activated protein kinase (MAPK) pathway, the PI3K, and c-Jun N-terminal kinases (JNKs) are subsequently initiated, which in turn can induce the proliferation and migration of angiogenic ECs. Integrin α_v_β_3_ is also known to work synergistically with VEGF to activate angiogenesis in ECs via VEGFR-2 phosphorylation [162,163,164]. Etaracizumab (vitaxin), a humanized mAb against α_v_β_3_ integrin receptor, is the only antagonist that has entered phase II clinical trials for RA treatment. Despite showing good clinical benefits, further investigations were halted, probably due to the side effects observed and the limited efficacy of anti-angiogenic factors alone in controlling the disease progression [162,163,165].

Considering the potential of integrins as therapeutic targets for the treatment of RA, pinpointing the role of integrins in angiogenesis has been subject of a multitude of studies. PEG-HM-3, a novel integrin inhibitor, displayed effective anti-rheumatic activity in antigen-induced arthritis (AIA) and CIA animal models either as single agents or in combination therapy with methotrexate [166]. Another study demonstrated that urokinase-type plasminogen activator receptor (uPAR) could alter the biological characteristics of RA fibroblast-like synoviocytes via the β_1_-integrin/PI3K/Akt signaling pathway and thereby affect the neo-angiogenesis of synovial tissues in RA patients [167]. α_v_β_3_, β_1_-integrins predominantly localize to vascular endothelium and lining layer cells in RA synovial tissue, in contrast to OA and normal control synovial tissue. Of note, acute-phase serum amyloid A (A-SAA) was found to increase α_v_β_3_ and β_1_ binding in RA synovial fibroblasts [168]. The role of IL-18 as an angiogenic mediator in RA was reported to act via α_v_β_3_ integrin and independently on the contribution of local TNF-α, as evidenced by neutralizing anti-TNF-α mAb in the in vivo Matrigel plug model [169]. High molecular weight kininogen (HK) is a plasma protein that is cleaved by plasma kallikrein, leading to the production of a kinin-free derivative of HK (i.e., HKa). The formation of HKa results in the exposure of its domain 5 (D5). HKa or D5 inhibit EC migration and proliferation. The mAb C11C1, which prevents the binding of HK to ECs, inhibited angiogenesis in experimental inflammatory arthritis [170]. In vitro, HKa or D5 abrogated EC adhesion to vitronectin and fibrinogen, causing anoikis and apoptosis. According to these findings, uPAR acting as a HKa receptor was considered to form a signaling complex containing α_v_β_3_ and α_5_β_1_ and their associated signaling intermediates. HKa competes with vitronectin to bind to uPAR and disrupts this complex. Consequently, the mAb C11C1 has been proposed as a novel therapeutic to control angiogenesis in inflammation [170].

Since discovering the RGD motif, multiple peptidic and non-peptidic RGD-based integrin ligands with various degrees of specificity have been developed. Despite their similarities, the integrins can recognize different RGD-containing ECM proteins and respond differently to the interaction with each one of them [162]. Molecular imaging through RGD-based ligands for the non-invasive assessment of angiogenesis is of great current interest [171].

^68^Ga-Aquibeprin and ^68^Ga-Avebetrin are two PET tracers that have been explored in the early detection of arthritis in CIA: ^68^Ga-Aquibeprin as an α_5_β_1_ integrin targeted trimeric pseudopeptide and ^68^Ga-Avebetrin as an α_v_β_3_ targeted cyclo (RDGfK) trimer [39,172]. Although both tracers detected arthritis disease activity, the α_5_β_1_ targeting tracer uptake was identified earlier than the α_v_β_3_ targeted tracer in RA development, possibly even before the development of clinical symptoms. Interestingly, through comparison of the intensity of the integrin PET signals over time and correlation with the clinical RA indicators (i.e., arthritis scores), it was noted that during the progression of arthritis in CIA, the α_5_β_1_-integrin was not only upregulated earlier than α_v_β_3_, but increased consistently, and elevated to a higher expression level than α_v_β_3_ at later stages. Furthermore, the selectivity and specificity of both tracers to their respective integrin targets were confirmed by blockade and cross-blockade experiments. Cross-blockade by the saturation of the non-target integrin for each radiotracer did not influence the PET signal intensity, indicating that the observed signal in PET scans is only due to the binding of the corresponding radiolabeled ligand [39].

The first-in-human study to investigate the use of integrin in molecular imaging for the detection of synovial angiogenesis in RA utilized a cyclic PEGylated RGD peptide chelated with NOTA and monitored the response to treatment in comparison to ^18^F-FDG [79]. In this study, high levels of α_v_β_3_ integrin expression were observed in IHC analysis in the vascular ECs of patients with active RA. ^68^Ga-PRGD_2_ PET/CT provided a better reflection of the disease severity and changes in the clinical measures than ^18^F-FDG in response to the therapeutic interventions. Additionally, the uptake pattern of the joints was different using ^68^Ga-PRGD_2_ imaging, where it displayed a diffuse synovial involvement of the affected joints and tendon sheaths of patients with RA, while the accumulation of ^68^Ga-PRGD_2_ was confined to a specific diseased area in patients with OA. Finally, unlike ^18^F-FDG, the integrin tracer did not accumulate in the axillary lymph nodes [79].

It has previously been reported that mast cells are the primary source of two important proangiogenic factors, IL-8 and TNF-α, in the inflamed synovial tissue. GPI-induced arthritis relies strongly on the presence of mast cells, but is largely independent of T and B cells. The activation of mast cells in this arthritis model has been found to activate α_v_β_3_ integrin during early angiogenesis and induce pericyte proliferation, which occurs during the differentiation of mature blood vessels. Angiogenesis quantification by in vivo measuring of the activated α_v_β_3_ integrin using ^18^F–galacto-RGD and IHC studies corroborated this finding [81].

The therapeutic effects of bevacizumab (Avastin), a humanized mAb against VEGF and the first FDA approved drug for tumor angiogenesis, have been explored by ^99m^Tc-3P4-RGD_2_ in a rat CIA model. The in vivo imaging studies using this SPECT tracer demonstrated reduced uptake after TNF and VEGF therapy. Of note, both treatments exhibited similar effects on amelioration of arthritis [87].

^99m^Tc-(HYNIC(tricine/TPPTS)-3PRGD_2_) is another RGD-based tracer that has been subjected to in vitro and in vivo evaluation for RA in a rat arthritis model. The radiolabeled tracer had renal excretion and was partly metabolized via the hepatobiliary route. Tracer uptake in the arthritic ankles was observed from 1 h onward with continuous visualization of arthritic ankles up to 6 h. The uptake of arthritic ankles of the rats in both planar imaging and biodistribution correlated well with the severity of arthritis. The linear relationship was also observed between the arthritic ankle uptake and the integrin α_v_β_3_ expression levels, suggesting the practicability of molecular imaging and the estimation of α_v_β_3_ expression for a selection of patients who may benefit from anti-angiogenesis therapy [83].

An in vivo SPECT imaging study with three different targets (i.e., FAP (^111^In-28H1), macrophages (^111^In-anti-F4/80-A3-1), and integrin α_v_β_3_ (^111^In-DOTA-E-(cRGDfK)_2_) was performed to monitor the response to anti-TNF therapy (etanercept) in arthritic mice. Even though the three tracers differed in their targets, mechanism, and pharmacokinetics, the effect of etanercept on the macroscopic/clinical arthritis score could be quantitatively measured and correlated with all of the tracer imaging profiles. However, the decreased target uptake by these radiotracers after treatment needs further confirmation [86]. Of note, the RGD peptide-based tracer had a shorter imaging acquisition time (at 1 h) compared to the other two tracers ^111^In-anti-F4/80-A3-1 at 24 h and ^111^In-28H1 at 48 h.

Aiming to develop a novel PET tracer with high PET signal in vivo, three RGD peptides conjugated with a multimeric bifunctional chelator (tris(hydroxypyridinone)), to incorporate more than one radioisotope, were evaluated in serum transfer-induced arthritis in mice. Based on in vitro uptake, in vivo biodistribution, and target tissue accumulation, one conjugate (^68^Ga(HP_3_-RGD_3_)) was selected for arthritis imaging in severe and mild arthritis. ^68^Ga(HP_3_-RGD_3_) did not show statistically significant differences in joint uptake between the healthy mice and mice with mild arthritis, but was able to visualize severe arthritis. The uptake of ^68^Ga(HP_3_-RGD_3_) in severe arthritis was comparable to the previously studied ^111^In-DOTA-E(cRGDfK)_2_ tracer in CIA [80].

^99m^Tc-3PRGD_2_ (^99m^Tc-maraciclatide) scintigraphy showed a significant increase in tracer uptake in the joints of arthritic rats versus the untreated controls, consistent with the expression levels of α_v_β_3_ and CD31 being higher in rat arthritic joint tissue compared to the controls. Bevacizumab therapy ameliorated the arthritis severity and reduced the radiotracer uptake in affected joints. This tracer was brought forward for testing in a patient with active RA where it proved its applicability in the early detection of synovial angiogenesis in RA. In contrast, bone scanning using ^99m^Tc-MDP failed to distinguish early RA joints from the healthy controls [84].

The ^99m^Tc-maraciclatide uptake in the joints of RA patients and its correlation with power Doppler (PD)US has also recently been studied. Both the PDUS and radiolabeled tracer results were in agreement with thee angiogenic markers and disease activity, however, planar imaging was considered to have the advantage of providing quantitative whole-body information with a single acquisition and easier interpretation compared to US/MRI. Further investigation in a larger study in RA patients has therefore been proposed [85].

The molecular imaging of angiogenesis by ^68^Ga-NODAGA-RGDyk and ^18^F-FDG was investigated in a patient with squamous cell carcinoma to detect the primary tumor and metastatic cervical lymph nodes. Interestingly, the uptake was also observed in the bilateral joints (the shoulders, elbows, wrists, metacarpophalangeal, interphalangeal, and hip joints) of the same patient, indicating moderately active RA. Indeed, this patient had previously been diagnosed with RA. Both tracers showed the same level of uptake in the arthritic joints, but the RGD peptide-based tracer exhibited a higher target-to-nontarget ratio [82].

Recently, a new diphosphine chemical platform (BMA or (diphosphine, 2,3-bis(diphenylphosphino)maleic anhydride)) enabling simple, one-step, kit-based ^99m^Tc-radiolabeling of receptor-targeted peptides, has been employed for the synthesis of a cyclic RGD peptide (Arg-Gly-Asp-DPhe-Lys (RGD)). The radiolabeled tracer compound ^99m^TcO_2_(DP-RGD)_2_ demonstrated a high metabolic stability and selective accumulation in the joints of arthritic mice, which was in accordance with the joint swelling. Unlike other existing RGD-based radiotracers investigated thus far in RA, this one step, kit-based protocol, together with its comparable ability for angiogenesis detection, seems to pave the way for its widespread application in future studies [88].

### 3.6. Cyclooxygenase Enzymes

Cyclooxygenases (COXs) are key enzymes in prostaglandin (PG) biosynthesis from arachidonic acid metabolism and constitute the main target for non-steroidal anti-inflammatory drugs (NSAIDs) [173,174]. Two distinct isoforms of COX are expressed in the synovium. COX-1 is constitutively expressed, particularly by synovial lining cells, with no difference in its expression between inflammatory and OA. COX-2 is expressed by ECs, mononuclear leukocytes, and fibroblasts, and its expression is increased in inflammatory arthritis compared to that in OA [118]. COX-2 enzyme activity is upregulated in the rheumatoid synovium, leading to elevated PG production [175]. Both Prostaglandin E1 and Prostaglandin E2 (PGE1 and PGE2) are potent mediators of angiogenesis [176]. Pro-inflammatory cytokines such as IL-1β and TNF-α are also implicated in the overexpression of COX-2 [177]. The blockade of COX-2 expression by RNA-interference in RA fibroblast like synoviocyte (FLS) markedly reduced the VEGF levels. Similarly, COX-2 inhibitors can inhibit VEGF expression in RA FLSs; in contrast, exogenous PGE2 disrupts this inhibitory effect [174]. Studies exploring the effect of COX inhibitors on arthritic animal models have established that part of their effectiveness is mediated by suppressing the VEGF signaling pathway [178,179,180,181].

Some non-steroidal anti-inflammatory drugs (NSAIDs) have been radiolabeled for monitoring COX-1/-2 in RA. Using a CIA rat model of arthritis, PET imaging for COX-1 and COX-2 was performed with ^11^C-ketoprofen before and after TNF-α treatment (etanercept). The ^11^C-ketoprofen PET images showed a time-dependent accumulation of the tracer in the inflamed joint according to the progression of arthritis, although there was a moderate correlation between the tracer uptake and the level of paw swelling. After treatment, along with the decrease in paw swelling, the joint uptake was also reduced [90], underscoring its ability to monitor the inflammatory status during treatment. On the other hand, the development of a PET tracer by radiolabeling two isomers of ibuprofen (the (S)-1 isomer inhibits COX activity and (R)-1 isomer is inactive) with carbon-11 showed an increased accumulation of both radiotracers in the arthritic joints of animals. Lack of a different accumulation level between the enantiomers indicates that ibuprofen retention is regardless of the expression of COX expression [89].

Another novel PET tracer selectively targeting COX-2, ^11^C-MC1, was capable of visualizing the actively inflamed joints of RA patients in accordance with the increased number and activity of leukocytes in the synovial membranes. A blocking study with celecoxib, a preferential COX-2 inhibitor, confirmed the specificity of ^11^C-MC1 binding to COX-2 [91].

### 3.7. Cytokines

Pro-inflammatory cytokines may exert either direct angiogenic activity or indirect activity via the VEGF-dependent pathways [16]. TNF-α is a pleiotropic cytokine exerting diverse functions: the activation of leukocytes, ECs, and synovial fibroblasts; the induction of cytokines, chemokines, adhesion molecules, and matrix enzymes; the suppression of regulatory T-cell function; the activation of osteoclasts and the resorption of cartilage and bone; and the mediation of metabolic and cognitive dysfunction [182]. TNF-α is also considered as a potent pro-angiogenic mediator with an efficacy comparable to fibroblast growth factors (FGFs) [179]. TNF-α itself promotes neovascularization and may also regulate capillary formation via the angiopoietin 1 and angiopoietin 2 (Ang1/Ang2) Tie2-VEGF network [16,178,183]. In CIA, TNF-α regulated Tie2 activation involving interactions between ECs and synoviocytes. TNF-α upregulated Tie2 expression in ECs in a nuclear factor kappa B dependent process and also upregulated Ang1 in synoviocytes [184,185]. TNF-α stimulates the production of MMPs and PGE2 through the activation of intracellular signaling pathways including mitogen-activated protein kinase (MAPK) and activator protein-1(AP-1) [186]. The TNF-α blockade by infliximab, administered as a single agent or in combination with methotrexate, reduced the VEGF expression and vascularity within the RA synovium [16,187,188]. Simultaneous targeting of the TNF and Ang2 with a bispecific antibody showed anti-arthritic efficacy in an in vivo model of arthritis [189]. In an ex vivo setting, neo-angiogenesis in rheumatoid synovitis resulted from, at least in part, the angiogenic effect of locally produced TNF-α and platelet-activating factor (PAF) [190]. Consistently, (pre)clinical TNF blocking therapy studies reduced the joint inflammation by decreasing the expression of adhesion molecules as key players in the angiogenesis process [191,192,193,194,195]. Moreover, several novel natural and synthetic drugs such as triptolide, evening primrose oil (EPO) rich in gamma linolenic acid (GLA), niclosamide, wen luo yin, etc. have been shown to inhibit TNF-α production along with the inhibition of angiogenesis in vitro and in vivo in arthritis models [192,196,197,198,199].

Some studies have evaluated TNF-α expression using radiolabeled monoclonal antibodies (e.g., adalimumab and infliximab) or antibody fragments (e.g., certolizumab pegol) to evaluate the inflammatory activity in rheumatoid arthritis.

Infliximab, a chimeric anti-TNF-α antibody first used for RA treatment, was coupled with ^99m^Tc and used to differentiate the responder to non-responder patients of anti-TNF-α therapy. The image analysis showed a trend of faster blood clearance due to the formation of infliximab–anti-infliximab complexes and higher liver/spleen uptake by the removal of the antibody complexes in a non-responding patient. Clinically inflamed joints also showed tracer uptake. Comparatively, another study using the same tracer confirmed the predictive role of ^99m^Tc-anti-TNF mAb in therapy decision-making for the selection of RA candidates for infliximab therapy. In contrast, radiolabeled human nonspecific immunoglobulin (HIG) was not able to distinguish the changes in disease activity [96,97].

Adalimumab (Humira), a fully human, immunoglobulin G1 (IgG1), anti-TNF mAbm has also been assessed as an imaging modality [200] that could confirm or exclude RA diagnosis and select patients with disease activity to initiate anti-TNF-α therapy. In one study, the concordance of MRI for RA detection and ^99m^Tc-adalimumab scintigraphy was strong. Furthermore, its performance in distinguishing between the inflammatory and non-inflammatory sites in the affected joints was reported to be of high accuracy and sensitivity, with specificity values around 90% [95].

Certolizumab pegol (CZP), another humanized mAb with a specificity for human TNF-α, has been investigated in different studies with different radiolabeling techniques for the imaging of patients with various types of inflammatory arthritis including RA. CZP differs from other TNF blockers due to its link with polyethylene glycol (PEG), which enhances the preferential distribution to inflamed tissue over normal tissue to a greater extent compared to adalimumab and infliximab [94,201]. These studies demonstrated the feasibility of immuno-PET imaging of CZP, which was radiolabeled with ^89^Zr or ^99m^Tc for the visualization of TNF-driven diseases in a preclinical setting or in patients with active RA. Blocking tests confirmed that the PET images corresponded to the localization of active clinical inflammation and the tracers’ specificity [92,93,94].

### 3.8. Other

#### 3.8.1. Microvascular Endothelium

A synovial endothelium targeting peptide (SyETP, CKSTHDRLC) has been identified to differentially locate at synovial xenografts in a human/SCID mouse chimeric model of RA, but not to the skin or mouse microvascular endothelium [202]. This peptide sequence maintained its affinity and specificity in vivo to bind to synovial microvascular ECs and has been proposed for use as a targeting carrier for selective drug delivery [203]. To this end, a (radiolabeled) novel fusion protein consisting of human IL-4 (anti-inflammatory cytokine) linked via a MMP-cleavable sequence to multiple copies of SyETP was evaluated for its anti-angiogenic potential in an in vivo SCID mouse model of arthritis along with imaging using nano-SPECT/CT. The iodine-125 labeled IL-4-SyETP accumulated in the synovial but not in the control skin xenografts [98].

#### 3.8.2. Discussion

Synovial inflammation, particularly the proliferative pannus in RA, relies on synovial angiogenesis, which fuels multiple processes contributing to sustained inflammation in the course of the disease, and major pro-angiogenic factors have been identified in synovial fluid, synovial tissue, and plasma/serum [204]. Despite the discovery of angiogenesis as a key driver in RA pathobiology [205] and the identification of a large number of vascular targets, its utility for diagnostic non-invasive PET or SPECT imaging and therapeutics is still in its infancy. The majority of RA-related studies conducted in this context were performed at a preclinical level and should now proceed to the clinical assessment of the synovial neo-vasculature and its response to treatment. The molecular imaging studies targeting angiogenesis in RA have mostly been compared with ^18^F-FDG PET and bone scan (MDP/HDP) as reference molecular imaging tools. However, since these radiotracers lack specificity for the angiogenic process, there is an unmet need to identify more superior tracers for this purpose.

The six most promising tracers for the imaging of angiogenesis in RA, as described in the reported clinical studies, target binding sites in the ECM or on vascular adhesion molecules such as VAP-1 and α_v_β_3_ and α_5_β_1_ integrins, which are upregulated in response to inflammation and growth factors. VAP-1 is a protein with a dual function as it acts both as an adhesion molecule and has amine oxidase enzymatic activity. Recently, the first in-human study with ^68^Ga-labeled siglec-9 as the VAP-1 ligand supported its candidature as a highly promising target to assess the angiogenic/inflammatory process in RA [73]. Other recent insights of sialylated siglec-9 being expressed on macrophages may add another role of VAP-1 in RA inflammatory programs [206].

Many efforts in integrin targeting have focused on α_v_β_3_ with all radio-ligands applied in human RA studies being RGD-based peptides. The small size of the peptides allows them to be rapidly distributed and excreted from the body. Several promising variants of these RGD peptides with improved biodistribution and pharmacokinetics for α_v_β_3_ integrin [79,84,207] are currently under clinical investigation for RA. Although α_v_β_3_ is the main integrin used for imaging and targeting purposes, with relatively successful clinical application in the context of oncology and even in RA, there is still room for exploring other members of the integrin family. In this respect, ongoing screening studies have identified α_5_β_1_-selective novel (bicyclic) RGD peptides for future evaluation [162,208,209].

Synovial hypoxia is a hallmark triggering angiogenesis and five tracers (three for PET, two for SPECT) showed the feasibility of visualizing this process in RA already in the early stages of the disease. The chemical properties of the tracers accounted for a higher target to background ratio for ^18^F-FAZA (hydrophilic nature) than ^18^F-FMISO (lipophilic nature), which may determine the best choice for hypoxia imaging in RA for one of these or alternatives such as ^64/67^Cu-ATSM [210]. Of future interest, recent developments in hypoxia imaging in oncology include the testing of galectin tracers. Galectins, particularly galectin-1, -3, and-9, are implicated in angiogenesis [211,212,213,214] and in recent research, ^68^Ga-galectracer was evaluated for the PET imaging of tumor hypoxia, displaying superiority over FMISO in the imaging of hypoxia in animal models and allowed for the early prediction of the tumor response to radiotherapy [215]. Thus, ^68^Ga-galectracer may also be a candidate for imaging hypoxia in RA.

Diagnostic relevance of the somatostatin receptor has been established in different types of inflammatory diseases including RA with SST2 overexpression by ECs [216]. Initial scintigraphy studies with somatostatin analogue tracers claimed differential tracer binding to vascular pannus in active rheumatoid joints over non-vascularized joints [67,110], but since a very recent study with a somatostatin analogue PET tracer (DOTANOC) did not show added value over the reference MDP scan to detect clinically positive joints for RA, the clinical utility of somatostatin tracers warrants further exploration [65].

PET or SPECT imaging of MMPs is still in a preliminary phase both in oncology and RA studies, with no clinical evaluation reported to date [46]. Recently, potent MMP-13 inhibitors have been successfully synthesized and radiolabeled with the PET radioisotope. Further modifications to the tracers were recommended to improve their pharmacokinetic properties, after which (pre)clinical arthritis models are awaited to reveal their potential for angiogenesis imaging in RA.

Dual (low-dose) tracer imaging is another emerging strategy that can improve the contrast and specificity by providing a larger potential of binding sites, and as such, may be beneficial in the field of RA angiogenesis when it is hard to define and employ a single target for imaging. The conjugate F8-IL10 showed initial encouraging imaging results in RA animal and human studies, but awaits further confirmation. Another candidate as a dual targeting tracer for angiogenesis may be ^68^Ga-NOTA-3P-TATE-RGD, which binds specifically to integrin and somatostain [217,218]. Following oncology, this tracer is eligible for evaluating the RA angiogenesis in preclinical models.

Not all of the reported tracers in this overview specifically target RA angiogenesis. This holds relevance when evaluating the treatment efficacy of a compound related to its angiogenesis inhibitory effects. COX and TNF-α inhibitors are regularly used in RA clinical practice to elicit anti-inflammatory effects, but they also confer the inhibition of EC migration and synovial angiogenesis [16]. Their associated tracers may be particularly helpful in identifying patients responding to these therapies, although both the imaging and the clinical effects of these treatments may be largely independent of angiogenesis.

Angiogenesis is an attractive target due to its integral role in various diseases. Given the overlapping mechanisms between tumor and RA angiogenesis, potentially useful tracers identified in oncology in terms of desirable specificity, pharmacokinetics, and targeting opportunities support the translation of encouraging research outcomes to the RA field. Beyond the examples indicated above, some new angiogenic targets derived from oncology studies include urokinase plasminogen activator receptor (uPAR). uPAR on the neo-angiogenic endothelium supports the infiltration of inflammatory cells. ^64^Cu/^68^Ga-DOTA-AE105 is a peptide-based uPAR tracer in the clinical phase of testing, which may also be considered for the diagnosis of RA angiogenesis activity in the early phase of the disease [219]. The sphingosine-1-phosphate (S1P) pathway may also be another interesting target to investigate in the context of angiogenesis therapy and imaging. This target is among a growing number of studies that have investigated its role in RA and malignancies [7,220,221,222,223,224,225]. Imaging modalities are becoming increasingly important for diagnostic and therapy response monitoring studies. Conceivably, imaging can also be positioned in areas such as the stratification of patients for personalized medicine treatment, prognosis and pathogenetic research. As such, angiogenesis imaging can contribute, to a certain degree, to the optimal diagnosis and treatment of RA patients.

The current review had some limitations. The parameters in the search strategy were chosen to find the angiogenic targets involved in RA and based on those to identify the molecular imaging markers (tracers) that have the potential to visualize angiogenesis in RA in clinical and preclinical phases of the disease. This search strategy could have led to some relevant studies being missed, especially in the preclinical stages of the disease as well as radiopharmaceuticals that have not been primarily tested or developed for RA. In addition, angiogenesis biomarkers possess multiple functions in RA and thus may have been missed if described under an alternative role. Finally, the restriction of the search strategy with three terms to identify angiogenesis targets could also have resulted in missing applicable targets.

## 4. Conclusions

This review provided a summary of the molecular imaging tracers with a potential applicability in the detection or monitoring of angiogenic activities as an important component of RA disease. Adhesion molecules particularly targeting VAP-1 and integrins, up to now, seem to hold promising results due to their critical roles in angiogenesis. Nonetheless, to identify the most appropriate angiogenic molecular target for imaging, further performance optimizations and/or clinical evaluations of the currently introduced radiotracers as well as the exploration of new targets are required to expand our insights into exploiting angiogenesis in RA. Overall, targeting angiogenesis by radiotracers holds great promise for the early diagnosis, selection of patients, and treatment evaluation in RA.

## Figures and Tables

**Figure 1 ijms-23-07071-f001:**
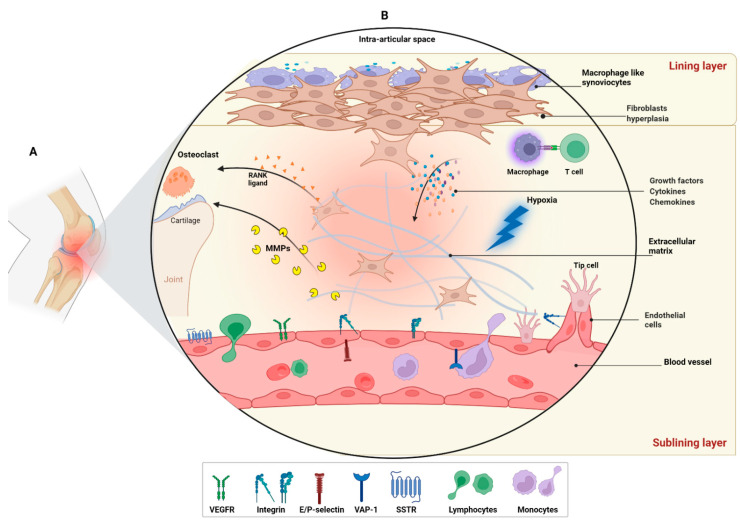
(**A**) Schematic view of an inflamed arthritic knee. (**B**) Detailed viewed of the inflamed RA synovium, particularly highlighting processes involved in angiogenesis.

**Figure 2 ijms-23-07071-f002:**
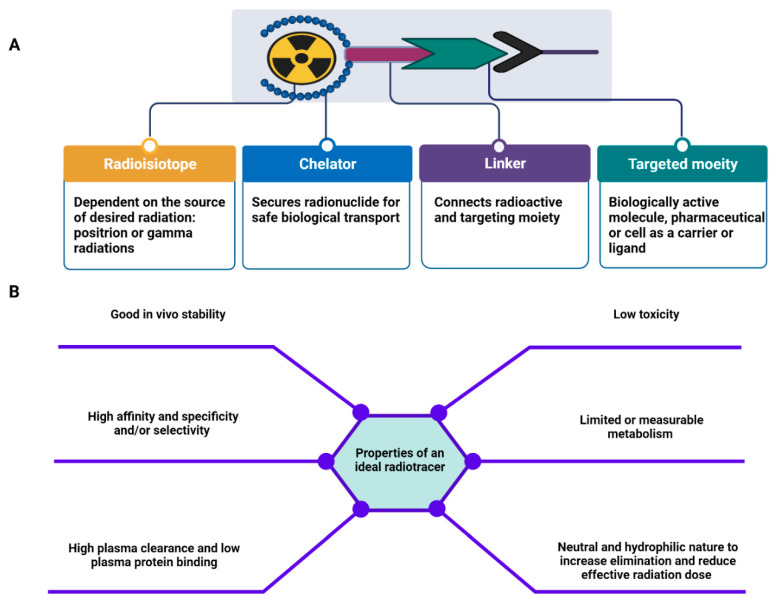
(**A**) The general structural components of a radiotracer. (**B**) Ideal radiotracer properties.

**Figure 3 ijms-23-07071-f003:**
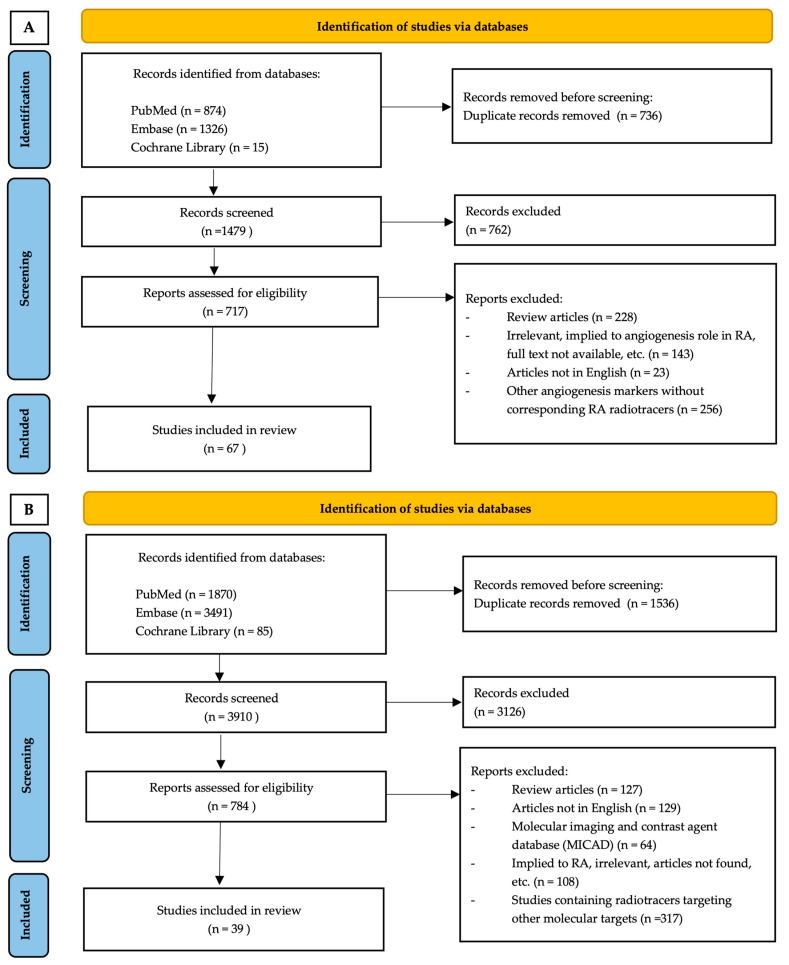
The flow diagram of the search and selection procedure of studies. (**A**) Overview of the articles found during the search for RA-angiogenesis targets. (**B**) Overview of the articles found during the search for nuclear imaging tracers in RA.

**Table 1 ijms-23-07071-t001:** Overview of all of the investigated PET/SPECT and scintigraphy tracers targeting angiogenesis in RA.

Type	Target	Radiotracer Name	Imaging Mode	Developmental Phase	Year	Ref.
** *Environmental* ** ** *factor* **	*Hypoxia*	^18^F-FMISO	PET	Animal	2017/2019	[60,61]
		^18^F-FAZA	PET	Animal	2017	[61]
		^67^Cu-ATSM	PET	Animal	2021	[62]
		α-[^123^I]-IAZA α-[^123^I]-IAZP	Scintigraphy	Human	2002/2001	[63] [64]
** *Somatostatin* **	*SST2*	^68^Ga- DOTANOC	PET	Human	2022	[65]
		^99m^Tc-HYNIC-TOC	Scintigraphy	Human	2016	[66]
		^111^In-DTPA-D-Phe-octreotide (pentetreotide)	Scintigraphy	Human	1994	[67]
		^99m^Tc-EDDA/HYNIC-TOC	Scintigraphy	Human	2012	[66]
		^99m^Tc-depreotide	Scintigraphy	Human	2017	[68]
** *Matrix Proteases* **	*MMP-2/-9*	^18^F-2,4,6-trione-30	PET	Animal	2012	[69]
	*MMP-13*	^11^C-MMP-13 inhibitor (2d)	PET	Animal *	2017	[70,71]
		^68^Ga-MMP13 inhibitor (2i)	PET	Animal *	2017	[70,71]
		^18^F-MMP-13 inhibitor (2a)	PET	Animal *	2017	[70,71]
** *ECM* **	*ED-A fibronectin*	^124^I-F8-IL10	PET	Animal/Human	2019	[72]
** *Adhesion molecules* **	*VAP-1*	^68^Ga-DOTA-Siglec-9	PET	Animal/Human	2021/2015	[73,74]
		^124^I-BTT-1023	SPECT	Animal	2013	[75]
	*E-selectin*	^111^In-anti-E-selectin	Scintigraphy	Animal	2009	[76]
		^99m^Tc-1.2B6-Fab	Scintigraphy	Human	1997/2002	[77,78]
	*Integrin a_5_B_1_*	^68^Ga-Aquibeprin	PET	Animal	2019	[39]
	*Integrin avB_3_*	^68^Ga-Avebretin	PET	Animal	2019	[39]
		^68^Ga-PRGD2	PET	Human	2014	[79]
		^68^Ga-HP(3)-RGD(3)	PET	Animal	2017	[80]
		^18^F-galacto-RGD	PET	Animal	2007	[81]
		^68^Ga-NODAGA-RGDyk	PET	Human	2021	[82]
		^99m^Tc-HYNIC-3PRGD_2_	Scintigraphy	Animal	2015	[83]
		^99m^Tc-3PRGD_2_	Scintigraphy	Animal/Human	2017	[84]
		^99m^Tc-maraciclatide	Scintigraphy	Human	2020	[85]
		^111^In-DOTA-E-[c(RGDfK)]_2_	SPECT	Animal	2016	[86]
		^99m^Tc-3P4-RGD_2_	SPECT	Animal	2013	[87]
		^99m^TcO_2_(DP-RGD)_2_	SPECT	Animal	2021	[88]
** *Enzymes* **	COX1–3	^11^C-Ibuprofen	PET	Animal	2011	[89]
		^11^C-Ketoprofen	PET	Animal	2017	[90]
	COX-2	^11^C-MC1	PET	Human	2020	[91]
** *Cytokines* **	TNF-α	^89^Zr-Certolizumab Pegol	PET	Animal	2020	[92]
		^99m^Tc- certolizumab pegol	Scintigraphy	Human	2016	[93]
		^99m^Tc-S-HYNIC-certolizumab pegol	Scintigraphy	Human	2016	[94]
		^99m^Tc-adalimumab	Scintigraphy	Human	2021	[95]
		^99m^Tc-Infliximab	Scintigraphy	Human	2007/2012	[96,97]
**Other**	*IL-4- ECs of inflamed tissue*	^125^I-IL-4-SyETP	SPECT	Animal	2013	[98]

* Not an arthritis animal model.

## Data Availability

Not applicable.

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
