# Peer review of "Systematic Review: Targeted Molecular Imaging of Angiogenesis and Its Mediators in Rheumatoid Arthritis"

_ijms, 2022, doi:10.3390/ijms23137071_

Round 1

Reviewer 1 Report

The manuscript titled, "Targeted molecular imaging of angiogenesis in rheumatoid arthritis", summarized imaging markers used for Single Photon Emission Computed Tomography (SPECT) and/or Positron Emission Tomography (PET) approaches, targeting pathways and mediators involved in synovial neo-angiogenesis in RA. This review article is fascinating and written well. However, one of the critical recent fundamental imaging approaches "second harmonic generation microscopy" which might be useful in the future is missing in the draft. For more information, check this recently published research article (https://doi.org/10.1038/s41598-022-13062-y).

Author Response

Response to Reviewer 1 Comment

The manuscript titled, "Targeted molecular imaging of angiogenesis in rheumatoid arthritis", summarized imaging markers used for Single Photon Emission Computed Tomography (SPECT) and/or Positron Emission Tomography (PET) approaches, targeting pathways and mediators involved in synovial neo-angiogenesis in RA. This review article is fascinating and written well.

Point 1: However, one of the critical recent fundamental imaging approaches "second harmonic generation microscopy" which might be useful in the future is missing in the draft. For more information, check this recently published research article (https://doi.org/10.1038/s41598-022-13062-y).

Response 1: This systematic review is focused on non-invasive in vivo nuclear imaging. Molecular imaging modalities, beyond nuclear imaging techniques, also consist of 2 other non-ionizing main categories; optical and acoustic (ultrasound) imaging with emerging powerful branches, as you mentioned, such as second harmonic generation microscopy for living cells and tissues. As you suggested, we addressed this point briefly in the introduction section and did not discuss it further in detail as it is beyond the scope of this paper.

Reviewer 2 Report

In the present review, the authors discuss nuclear imaging modalities as valuable non-invasive tools for RA and quantitatively track molecular changes in multiple joints. This systematic review summarizes imaging markers, using Single Photon Emission Computed Tomography (SPECT) and/or Positron Emission Tomography (PET), targeting pathways and mediators involved in the regulation of synovial pathophysiology in RA.

Overall, it is an interesting and comprehensive review. It is well-written and well-structured. Figures and tables help the reader to better follow the text.

Points to address

-Many of the markers that have been discussed do no strictly relate to angiogenesis. I recommend to modify the title and the text, to broaden its content beside angiogenesis

-Please include comments on Figure 2A-B in lines 136-137

-Suppl. material is missing: it is important to know the search strategy

Author Response

Response to Reviewer 2 Comments

In the present review, the authors discuss nuclear imaging modalities as valuable non-invasive tools for RA and quantitatively track molecular changes in multiple joints. This systematic review summarizes imaging markers, using Single Photon Emission Computed Tomography (SPECT) and/or Positron Emission Tomography (PET), targeting pathways and mediators involved in the regulation of synovial pathophysiology in RA. Overall, it is an interesting and comprehensive review. It is well-written and well-structured. Figures and tables help the reader to better follow the text.

Point 1: Many of the markers that have been discussed do no strictly relate to angiogenesis. I recommend to modify the title and the text, to broaden its content beside angiogenesis.

Response 1: We agree with the reviewer that not only direct targets of angiogenesis, but also targets of the mediators (direct or more indirect) were enclosed. Therefore, we changed the title to: ‘Targeted molecular imaging of angiogenesis and its mediators in rheumatoid arthritis’.

Point 2: Please include comments on Figure 2A-B in lines 136-137

Response 2: Comments are added to the main text in lines 136-137 for Figure 2A-B.

Point 3: Suppl. material is missing: it is important to know the search strategy

Response 3: Supplementary materials on search strategy are uploaded to the journal submission system.